# Adipocyte ALK7 links nutrient overload to catecholamine resistance in obesity

Tingqing Guo[1,2], Patricia Marmol[1], Annalena Moliner[1,3], Marie Björnholm[4], Chao Zhang[5†], Kevan M Shokat[5], Carlos F Ibanez[1,2,3]*

[1]Department of Neuroscience, Karolinska Institutet, Stockholm, Sweden; [2]Department of Physiology, National University of Singapore, Singapore, Singapore; [3]Life Sciences Institute, National University of Singapore, Singapore, Singapore; [4]Department of Molecular Medicine and Surgery, Section for Integrative Physiology, Karolinska Institutet, Stockholm, Sweden; [5]Department of Cellular and Molecular Pharmacology, Howard Hughes Medical Institute, University of California, San Francisco, San Francisco, United States

*For correspondence: carlos.ibanez@ki.se

Present address: †Department of Chemistry, University of Southern California, Los Angeles, United States

Competing interests: The authors declare that no competing interests exist.

**Abstract** Obesity is associated with blunted β-adrenoreceptor (β-AR)-mediated lipolysis and lipid oxidation in adipose tissue, but the mechanisms linking nutrient overload to catecholamine resistance are poorly understood. We report that targeted disruption of TGF-β superfamily receptor ALK7 alleviates diet-induced catecholamine resistance in adipose tissue, thereby reducing obesity in mice. Global and fat-specific *Alk7* knock-out enhanced adipose β-AR expression, β-adrenergic signaling, mitochondrial biogenesis, lipid oxidation, and lipolysis under a high fat diet, leading to elevated energy expenditure, decreased fat mass, and resistance to diet-induced obesity. Conversely, activation of ALK7 reduced β-AR-mediated signaling and lipolysis cell-autonomously in both mouse and human adipocytes. Acute inhibition of ALK7 in adult mice by a chemical-genetic approach reduced diet-induced weight gain, fat accumulation, and adipocyte size, and enhanced adipocyte lipolysis and β-adrenergic signaling. We propose that ALK7 signaling contributes to diet-induced catecholamine resistance in adipose tissue, and suggest that ALK7 inhibitors may have therapeutic value in human obesity.

## Introduction

Agonists of β3-adrenoceptors (β3-AR) are effective anti-obesity agents in rodents due to their ability to stimulate lipolysis and lipid oxidation in adipose tissue (*Arch, 2011*). However, efforts to develop similar compounds in humans stalled when it became clear that the pharmacology of rodent and human β3-AR differed and lipolysis in human adipocytes was mainly mediated by classical β1- and β2-AR, which can also induce hypertension, tachycardia, and other undesired complications (*Arch, 2011*). Alternative strategies to enhance catecholamine sensitivity and β-adrenergic signaling selectively in adipose tissue may be effective in combating obesity, but remain unproven. In both humans and rodents, obesity is associated with blunted β-AR-mediated lipolysis and lipid oxidation in adipose tissue (*Reynisdottir et al., 1994*; *Arner, 1999*; *Jocken et al., 2008*), but the mechanisms linking nutrient overload to catecholamine resistance remain poorly understood.

ALK7 is a receptor for a subset of ligands from the TGF-β superfamily, including Nodal, activin B, and GDF-3 (*Rydén et al., 1996*; *Reissmann et al., 2001*; *Tsuchida et al., 2004*; *Andersson et al., 2008*). ALK7 is not necessary for mouse embryogenesis (*Jörnvall et al., 2004*), suggesting alternative functions in postnatal development and adult physiology. ALK7 is highly expressed in rodent and human adipose tissue (*Kang and Reddi, 1996*; *Andersson et al., 2008*; *Carlsson et al., 2009*; *Murakami et al., 2012*), as well as in a few other tissues implicated in metabolic regulation, such as

**eLife digest** Adrenaline and noradrenaline are two hormones that trigger the burst of energy and increase in heart rate and blood pressure that are needed for the 'fight-or-flight' response. Both belong to a group of chemicals called catecholamines. These chemicals bind to cells carrying proteins called adrenoceptors on their surface and stimulate the breakdown of fat, which releases energy. However, when nutrients are plentiful, fat cells become resistant to catecholamines and instead store fat so it can be used for energy if food becomes scarce. In the industrialized world where food is easily and constantly accessible, this resistance can cause an unhealthy increase in body fat and result in obesity.

Increasing fat metabolism by making fat cells more able to respond to catecholamines is an attractive strategy for combating obesity. Indeed, drugs that mimic the effect of catecholamines on an adrenoceptor found in mice reduce obesity caused by over-eating. However, these drugs are ineffective in humans and can cause harmful side effects to the cardiovascular system, including high blood pressure and an increased heart rate. Devising a strategy that specifically targets catecholamine resistance in fat cells is therefore desirable.

A protein called ALK7 is a cell surface receptor that is predominantly found in fat cells and tissues involved in controlling the metabolism. Mice with a mutation in ALK7 that stops this protein from working properly accumulate less fat than mice with a functional version of the protein, but it is not known why. To understand ALK7's involvement in fat metabolism, Guo et al. created mice whose fat cells lack ALK7, but whose other cells all produce ALK7 as normal. When fed a diet rich in fat, these mice are leaner than regular mice and they burn more energy.

The metabolic responses seen in ALK7 mutant mice are very similar to those seen in mice treated with drugs targeting adrenoceptors, suggesting that there may be a link between ALK7 and catecholamine resistance. Indeed, Guo et al. demonstrate that fat cells lacking ALK7 have an increased sensitivity to catecholamines when the mice are on a high fat diet, which decreases the amount of fat the mice accumulate. Conversely, increasing the activity of ALK7 reduces the ability of the cells to respond to catecholamines, and they accumulate more fat.

Guo et al. also generated a second line of mice carrying a mutation in ALK7 that does not affect its function, but renders it sensitive to inhibition by a custom-made chemical. When these animals were on a high-fat diet, administering the chemical made the mice leaner, suggesting that inhibiting the ALK7 receptor can prevent obesity in adult animals.

Guo et al. also performed experiments in human fat cells, which showed that the ALK7 receptor works in a similar way in human cells as it does in mice. As ALK7 is largely specific for fat cells and is not known to affect the cardiovascular system, drugs that inhibit ALK7 could potentially safely suppress catecholamine resistance and reduce human obesity.

pancreatic islets (*Bertolino et al., 2008*) and the arcuate nucleus of the hypothalamus (*Sandoval-Guzmán et al., 2012*). Mice lacking ALK7 show reduced fat accumulation after a high fat diet (*Andersson et al., 2008*) and in a polygenic model of obesity (*Yogosawa et al., 2013*), but the mechanisms underlying the effects of ALK7 signaling on diet-induced obesity are not understood. Pancreatic islets from *Alk7* knock-out mice show enhanced glucose-stimulated insulin secretion (*Bertolino et al., 2008*), a phenotype that is also present in islets from mutant mice lacking the ALK7 ligand activin B (*Wu et al., 2014*). Moreover, the arcuate nucleus of *Alk7* knock-out mice shows reduced expression of *Npy* mRNA and lower numbers of *Npy*-expressing neurons compared to wild type controls (*Sandoval-Guzmán et al., 2012*). It has therefore been unclear whether ALK7 affects fat accumulation cell-autonomously in adipose tissue or through other sites, and whether its effects on adult physiology are developmental or homeostatic, via acute regulation of adult cell function.

In this study, we developed a conditional knock-out mouse lacking ALK7 in adipose tissue and a knock-in mouse model carrying an analogue-sensitive kinase allele (ASKA) of ALK7, which can be specifically inhibited by administration of ATP competitive inhibitors. Using these animals, as well as cell culture models, we have established that ALK7 functions cell-autonomously and acutely in adult adipocytes to control energy expenditure and fat accumulation by suppressing adipocyte mitochondrial biogenesis, fatty acid oxidation, and β-AR mediated-lipolysis. Importantly, we found that ALK7

signaling negatively regulates adipocyte β-AR expression and β-adrenergic signaling during a high fat diet, providing a link between nutrient overload and catecholamine resistance in adipose tissue.

## Results

### Fat-specific disruption of ALK7 signaling attenuates weight gain and fat accumulation under a high fat diet

In order to dissect the cell-autonomous functions of ALK7 in specific tissues, we generated a conditional knock-out allele of the mouse *Alk7* gene (also known as *Acvr1c*) with *loxP* sites flanking exons 5 and 6, encoding essential regions of the ALK7 kinase domain (*Figure 1—figure supplement 1*). Gene deletion in adipose tissue was achieved by crossing *Alk7*fx mice with *Ap2*CRE mice (*He et al., 2003*). Although *Ap2*CRE has also been reported to be expressed in adipose tissue macrophages (*Lee et al., 2013*), *Alk7* mRNA expression could only be detected in the adipocyte fraction of adipose tissue but not in the stromal-vascular fraction (containing macrophages) or in spleen (*Figure 1—figure supplement 2A–D*). Expression of *Alk7* mRNA was reduced by 60% in the adipose tissue of *Alk7*fx/fx::*Ap2*CRE mice, while a 98% reduction was achieved in *Alk7*fx/−::*Ap2*CRE mice (i.e., compound heterozygotes carrying floxed and knock-out *Alk7* alleles) (*Figure 1—figure supplement 3A,B*). No change in *Alk7* mRNA expression was observed in the pancreas or brain (*Figure 1—figure supplement 2B*). Both lines of fat-specific *Alk7* mutant mice showed significantly reduced weight gain during 12 weeks on a high fat diet compared to controls (*Figure 1A,B*). In contrast, weight gain in *Alk7*fx/fx::*Nestin*CRE mice, lacking ALK7 expression in the nervous system (*Figure 1—figure supplement 3C*), did not differ from controls (*Figure 1C*). Diet-induced fat accumulation, as measured by epididymal and retroperitoneal fat depot weight (*Figure 1D–F*), magnetic resonance imaging (MRI) of total fat mass (*Figure 1G,H*), and adipocyte cell size (*Figure 1I,J*), was also significantly reduced in fat-specific *Alk7* mutant mice compared to controls. In contrast, fat depots of nervous system-specific *Alk7* mutant mice were not different from controls (*Figure 1K*). In agreement with reduced diet-induced obesity, serum leptin levels were also lower after a high fat diet in both global and fat-specific *Alk7* knock-out mice (*Figure 2A,B*). However, fed serum insulin levels remained unchanged in fat-specific and brain-specific *Alk7* knock-out mice (*Figure 2C,D*), suggesting unaltered peripheral insulin sensitivity. In addition, glucose and insulin tolerance tests performed in fat-specific *Alk7* mutant mice and controls indicated normal glucose and insulin responses in the mutants (*Figure 2E–H*). Obesity has been associated with a state of inflammation in adipose tissue in which resident macrophages play important roles (*Hotamisligil, 2006*; *Fujisaka et al., 2009*). Following 8 weeks of a high fat diet, adipose tissue of global and fat-specific *Alk7* knock-out mice showed decreased expression of markers of pro-inflammatory M1 macrophages, such as *TNF1α, IL-12b*, and *Itgax* (*Figure 2I,J*), but increased expression of *Mgl2*, a marker of M2 macrophages, the major resident macrophages involved in remodeling and repair that are normally present in adipose tissue from lean mice (*Figure 2K,L*). This profile is in agreement with reduced adipose tissue inflammation and protection against diet-induced obesity. Together, these results indicate that ALK7 functions cell-autonomously in adipose tissue to regulate fat accumulation during nutrient overload.

### Increased energy expenditure and adipose tissue mitochondrial biogenesis in fat-specific *Alk7* knock-out mice on a high fat diet

The reduced obesity in *Alk7* knock-out mice after a high fat diet could be a result of lower calorie intake or higher energy expenditure. Both global knock-out and fat-specific *Alk7* mutant mice displayed increased energy expenditure (*Figure 3A,B*) and oxygen consumption (*Figure 3C,D*) after a high fat diet compared to controls. Food intake remained unchanged in the mutant mice (*Figure 3E*). Changes in energy expenditure in *Alk7* mutant mice were not due to 'browning' of subcutaneous adipose tissue, as expression of brown adipose tissue (BAT) marker genes *Ucp1* and *Elovl3* was not increased in the subcutaneous fat of the mutants (*Figure 3—figure supplement 1A,B*). Moreover, the browning effects of the β₃-AR-specific agonist CL316243 were comparable in subcutaneous adipose tissue of wild type and *Alk7* knock-out mice (*Figure 3—figure supplement 1C,D*). Neither was expression of BAT markers elevated in the BAT of *Alk7* mutant mice (data not shown). Global and fat-specific *Alk7* knock-out mice showed higher physical activity than wild type controls after a high fat diet (*Figure 3F,G*). However, it was recently reported that changes in activity do not drive changes in energy expenditure in groups of mice below thermoneutrality (*Virtue et al., 2012*). We hypothesized that increased energy expenditure in *Alk7* mutant mice on a high fat diet may be due to higher basal metabolic rate,

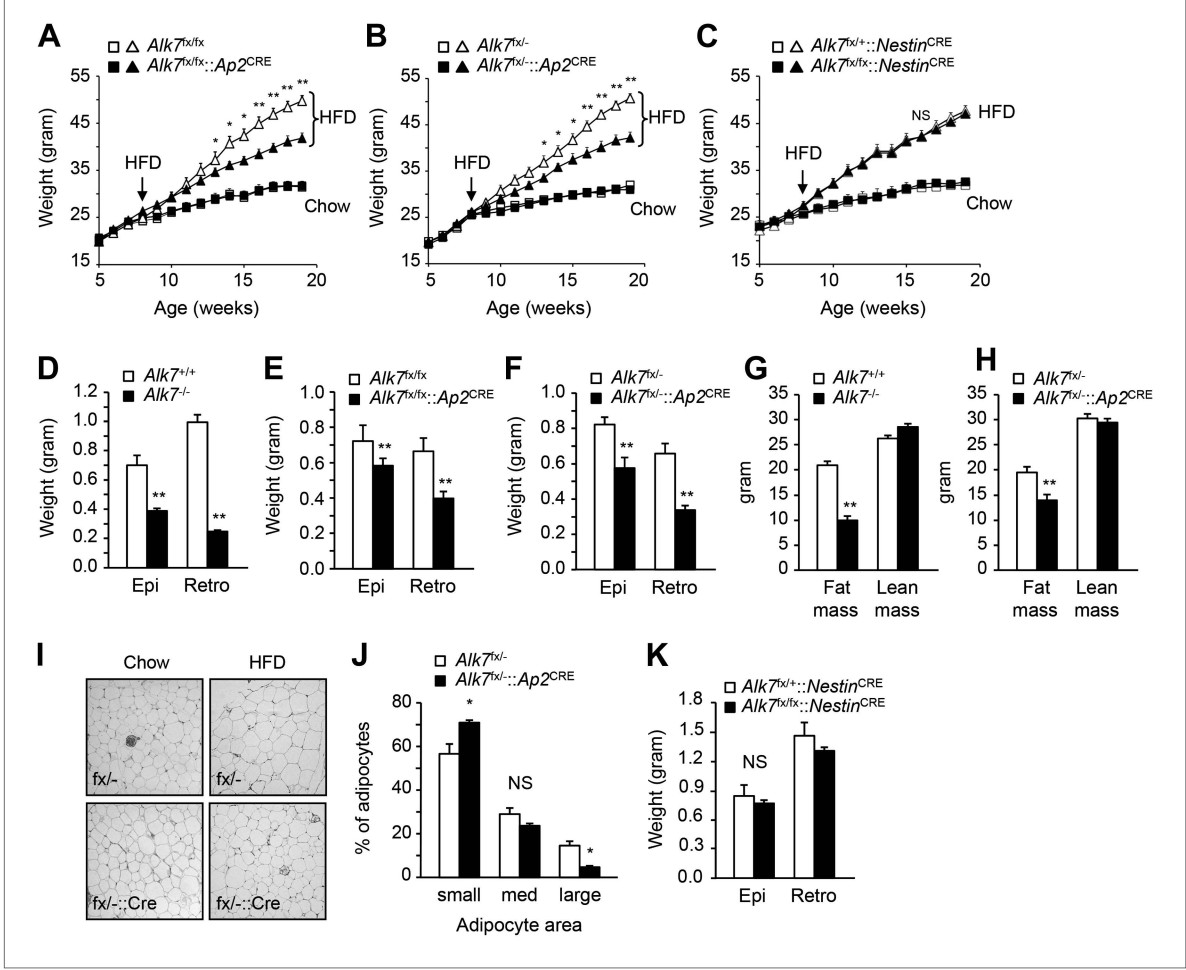

**Figure 1**. Conditional deletion of ALK7 in adipose tissue attenuates weight gain and fat deposition under a high fat diet. (**A–C**) Weight gain under chow (squares) and a high fat diet (HFD, triangles) in fat-specific $Alk7^{fx/fx}::Ap2^{CRE}$ (**A**) and $Alk7^{fx/-}::Ap2^{CRE}$ (**B**), and brain-specific $Alk7^{fx/fx}::Nestin^{CRE}$ (**C**) knock-out mice (solid symbols) compared to control mice (open symbols). Arrows denote the first week of HFD. N = 8 mice per group in all cases, except N = 6 in (**C**) on a chow diet. (**D–F**) Weights of epididymal (Epi) and retroperitoneal (Retro) fat depots in global $Alk7^{-/-}$ (**D**) and fat-specific $Alk7^{fx/fx}::Ap2^{CRE}$ (**E**) and $Alk7^{fx/-}::Ap2^{CRE}$ (**F**) knock-out mice after 16 weeks on HFD. N = 6 mice per group. (**G** and **H**) Fat and lean mass assessed by magnetic resonance imaging (MRI) in global $Alk7^{-/-}$ (**G**) and fat-specific $Alk7^{fx/-}::Ap2^{CRE}$ (**H**) knock-out mice after 16 weeks on HFD. N = 8 mice per group in (**G**), N = 5 in (**H**). (**I** and **J**) Adipocyte cell size in fat-specific $Alk7^{fx/-}::Ap2^{CRE}$ knock-out mice and $Alk7^{fx/-}$ controls after chow or HFD as visualized by hematoxylin-eosin staining in tissue sections of epididymal adipose tissue (**I**). Quantitative analysis is shown in (**J**). Small, 400–5000 μm²; Med, 5000–10,000 μm²; Large, 10,000–60,000 μm². N = 4 mice per group (four sections per mouse). (**K**) Weights of epididymal (Epi) and retroperitoneal (Retro) fat depots in brain-specific $Alk7^{fx/fx}::Nestin^{CRE}$ knock-out mice after 16 weeks on HFD. N = 6 mice per group. *p < 0.05; **p < 0.01; NS, non-significant (mutant vs control). All error bars show mean ± SEM.

The following figure supplements are available for figure 1:

**Figure supplement 1**. Generation of a conditional allele of the mouse *Acvr1c* gene encoding ALK7.

**Figure supplement 2**. *Alk7* expression in adipocytes, but not in adipose tissue macrophages.

**Figure supplement 3**. *Alk7* expression in conditional knock-out mice.

and investigated adipose tissue mitochondria biogenesis and function, which are impaired by chronic nutrient overload in rodents and humans (*Heilbronn et al., 2007*; *Rong et al., 2007*; *Sutherland et al., 2008*). Mitochondria biogenesis, as measured by mitochondrial DNA content (*Figure 4A,B*), citrate synthase activity (*Figure 4C,D*), and ATP content (*Figure 4E,F*), was significantly increased in adipose tissue of both global and fat-specific *Alk7* knock-out mice on a high fat diet compared to controls. In addition, several markers of mitochondrial biogenesis and function were also significantly

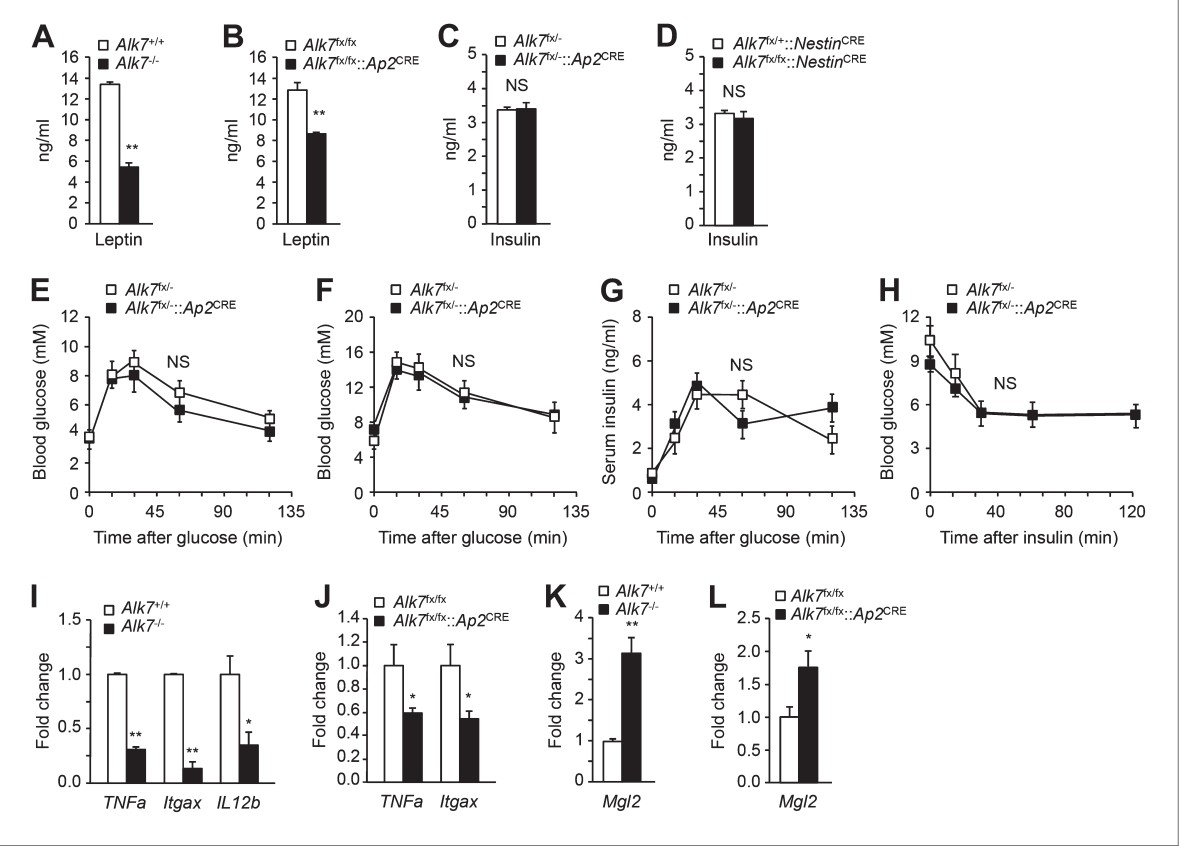

**Figure 2**. Reduced serum leptin levels and adipose tissue inflammation but normal glucose and insulin metabolism in global and fat-specific *Alk7* knock-out mice. (**A** and **B**) Serum levels of leptin in global *Alk7$^{-/-}$* (**A**) and fat-specific *Alk7$^{fx/fx}$::Ap2$^{CRE}$* (**B**) knock-out mice after 16 weeks on a high fat diet (HFD). N = 6 mice per group. (**C** and **D**) Serum levels of insulin in fat-specific *Alk7$^{fx/-}$::Ap2$^{CRE}$* (**C**) and brain-specific *Alk7$^{fx/fx}$::Nestin$^{CRE}$* (**D**) knock-out mice after 16 weeks on HFD. N = 6 mice per group. (**E** and **F**) Blood glucose levels after glucose injection in 4-month-old *Alk7$^{fx/-}$::Ap2$^{CRE}$* mice and controls after chow (**E**) or a high fat diet (**F**). Mice had been starved overnight prior to glucose injection. N = 5 mice per group. (**G**) Serum insulin levels after glucose injection in 4-month-old *Alk7$^{fx/-}$::Ap2$^{CRE}$* mice and controls kept on a chow diet. Mice had been starved overnight prior to glucose injection. N = 5 mice per group. (**H**) Blood glucose levels after insulin injection in 4-month-old *Alk7$^{fx/-}$::Ap2$^{CRE}$* mice and controls kept on a chow diet. Mice had been starved for 3 hr prior to insulin injection. N = 5 mice per group. (**I–L**) Relative mRNA expression levels of M1 macrophage markers *TNF1a*, *Itgax*, and *IL12b* (**I** and **J**) and the M2 macrophage marker *Mgl2* (**K** and **L**) assessed by quantitative PCR in epididymal adipose tissue of global *Alk7$^{-/-}$* (**I** and **K**) and fat-specific *Alk7$^{fx/fx}$::Ap2$^{CRE}$* (**J** and **L**) knock-out mice after 16 weeks on HFD. N = 6 mice per group. *p < 0.05; **p < 0.01; NS, non-significant (mutant vs control). All error bars show mean ± SEM.

upregulated in the mutants, including PGC-1α, a master regulator of mitochondrial biogenesis (***Wu et al., 1999***), mitochondrial uncoupling protein 3 (UCP-3), *cytochrome C*, and Hadhb, a key mitochondrial enzyme for β-oxidation of long chain fatty acids (***Middleton, 1994***) (***Figure 4G,H***). In line with these changes, fatty acid oxidation activity was enhanced in the adipose tissue of global and fat-specific *Alk7* mutant mice (***Figure 4I,J***). These results suggest that resistance to diet-induced obesity in *Alk7* mutant mice is due to increased energy expenditure and enhanced mitochondrial function and lipid oxidation in adipose tissue.

### Enhanced catecholamine sensitivity and β-adrenergic signaling in adipose tissue of global and fat-specific *Alk7* knock-out mice on a high fat diet

Activation of β-ARs by catecholamines is the major regulatory pathway of fat mobilization during starvation and exercise. Mice lacking all three types of β-ARs are massively obese on a high fat diet without an increase in food intake (***Bachman et al., 2002***). Conversely, β-agonist treatment induces mitochondria biogenesis in adipose tissue and reduces fat mass (***Ghorbani et al., 1997***), resembling the phenotype observed in *Alk7* mutant mice. β-AR-dependent fat mobilization is severely impaired during a high fat diet, allowing fat accumulation in adipose tissue, but the underlying mechanisms of

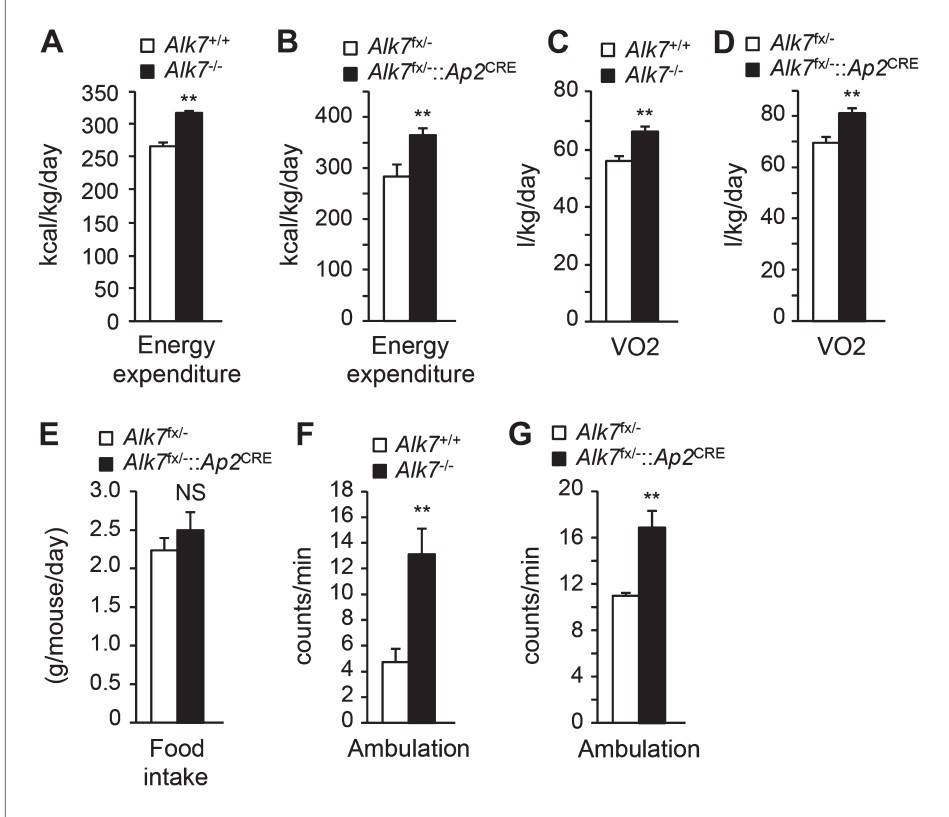

**Figure 3**. Increased energy expenditure and oxygen consumption in global and fat-specific *Alk7* knock-out mice on a high fat diet. (**A** and **B**) Energy expenditure assessed by calorimetric measurements in global *Alk7*−/− (**A**) and fat-specific *Alk7*fx/−::*Ap2*CRE (**B**) knock-out mice after 16 weeks on a high fat diet (HFD). N = 8 mice per group in (**A**), N = 6 in (**B**). (**C** and **D**) Oxygen consumption assessed by calorimetric measurements in global *Alk7*−/− (**C**) and fat-specific *Alk7*fx/−::*Ap2*CRE (**D**) knock-out mice after 16 weeks on HFD. N = 8 mice per group in (**C**), N = 6 in (**D**). (**E**) Daily food intake in fat-specific *Alk7*fx/−::*Ap2*CRE knock-out mice during 16 weeks on HFD. N = 6 mice per group. (**F** and **G**) Physical activity assessed as ambulation in global *Alk7*−/− (**F**) and fat-specific *Alk7*fx/−::*Ap2*CRE (**G**) knock-out mice after 16 weeks on HFD. N = 8 mice per group in (**F**), N = 6 in (**G**). *p < 0.05; **p < 0.01; NS, non-significant (mutant vs control). All error bars show mean ± SEM.

The following figure supplement is available for figure 3:

**Figure supplement 1**. *Alk7* deletion does not result in enhanced 'browning' of subcutanous adipose tissue.

catecholamine resistance have not been clarified (*Reynisdottir et al., 1994*; *Jocken et al., 2008*). We investigated whether ALK7 could play a role in the catecholamine sensitivity of adipose tissue under a low caloric diet and after nutrient overload. Under a chow diet, injection of the β₃-AR-specific agonist CL316243 enhanced lipolysis, as measured by free fatty acid release, significantly more in fat-specific *Alk7* knock-out mice than in wild type controls (*Figure 5A*). After 16 weeks on a high fat diet, global and fat-specific *Alk7* knock-out mice also displayed elevated lipolysis compared to controls under both basal conditions and following β₃-AR stimulation (*Figure 5B,C*). On the other hand, we detected no change in basal or agonist-stimulated lipolysis in brain-specific *Alk7* knock-out mice (*Figure 5D*). As levels of epinephrine and norepinephrine were not significantly altered in the adipose tissue of *Alk7* knock-out mice (*Figure 5—figure supplement 1*), these data suggested enhanced cat-echolamine sensitivity in adipose tissue lacking ALK7. In agreement with this, the levels of *Adrb1* and *Adrb3* mRNAs were significantly protected in *Alk7* knock-out mice after 16 weeks on a high fat diet, but were barely detectable in the adipose tissue of control mice (*Figure 5E,F*). In addition, expression of the negative regulator of adrenergic signaling *Rgs2*, an inhibitor of adenylate cyclase, was elevated after a high fat diet in the adipose tissue of wild type mice but not in *Alk7* knock-out mice (*Figure 5G*) or fat-specific *Alk7* knock-out mice (*Figure 5H,I*). *Adrb1* and *Adrb3* mRNA levels were also significantly

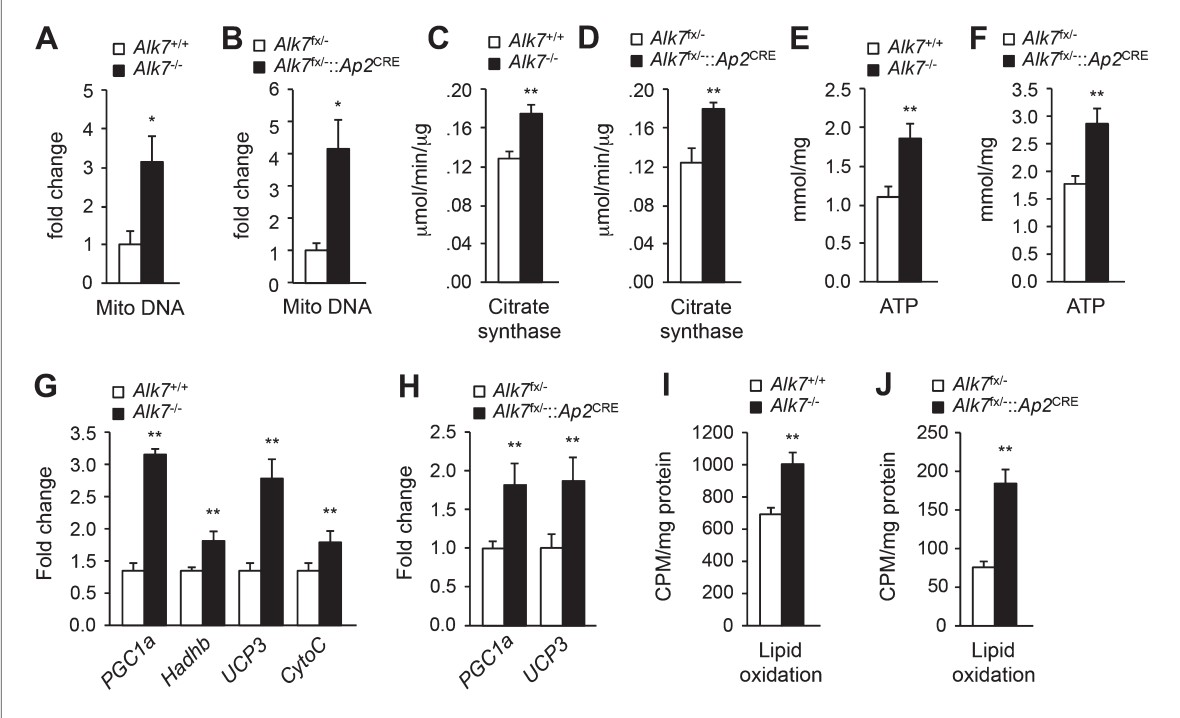

**Figure 4.** Elevated adipose tissue mitochondrial biogenesis and activity in fat-specific *Alk7* knock-out mice on a high fat diet. (**A–F**) Mitochondrial biogenesis assessed by measurements of mitochondrial (mito) DNA content (**A** and **B**), citrate synthase activity (**C** and **D**), and ATP content (**E** and **F**) in epididymal adipose tissue of global *Alk7*$^{-/-}$ (**A**, **C**, **E**) and fat-specific *Alk7*$^{fx/-}$::*Ap2*$^{CRE}$ (**B**, **D**, **F**) knock-out mice after 16 weeks on a high fat diet (HFD). N = 8 mice per group in all cases. (**G** and **H**) Relative mRNA expression of markers of mitochondrial biogenesis and function *PGC1a*, *Hadhb*, *UCP3*, and *cytochrome C* assessed by quantitative PCR (Q-PCR) in epididymal adipose tissue of global *Alk7*$^{-/-}$ (**G**) and fat-specific *Alk7*$^{fx/-}$::*Ap2*$^{CRE}$ (**H**) knock-out mice after 16 weeks on HFD. N = 6 mice per group in all cases. (**I** and **J**) Lipid oxidation in epididymal adipose tissue of global *Alk7*$^{-/-}$ (**I**) and fat-specific *Alk7*$^{fx/-}$::*Ap2*$^{CRE}$ (**J**) knock-out mice. N = 3 mice per group in all cases. *$p < 0.05$; **$p < 0.01$; NS, non-significant (mutant vs control). All error bars show mean ± SEM.

higher in fat-specific *Alk7* knock-outs compared to controls after a high fat diet (***Figure 5H,I***). In line with enhanced adrenergic signaling, we observed increased levels of total and phosphorylated hormone-sensitive lipase (HSL), a major target of the β-adrenergic pathway in adipocytes, and phosphorylated PKA substrates in the adipose tissue of *Alk7* knock-out mice after 16 weeks on a high fat diet compared to wild type controls (***Figure 5J,K***). Moreover, norepinephrine injection induced a more pronounced increase in PKA activity in the adipose tissue of *Alk7* knock-out mice than in control mice (***Figure 5L***). Together, these data suggest that cell-autonomous ALK7 signaling in adipose tissue suppresses catecholamine sensitivity and β-adrenergic signaling under a high fat diet.

## ALK7 signaling negatively regulates catecholamine sensitivity and β-adrenergic signaling in mouse and human adipocytes

The enhanced catecholamine sensitivity of adipose tissue in *Alk7* mutant mice prompted us to investigate the acute effects of ALK7 signaling on β-AR expression and adrenergic signaling in cultured adipocyte cells derived by differentiation from mouse embryonic fibroblasts (MEFs). Stimulation of wild type adipocytes with the ALK7 ligand activin B reduced expression of *Adrb2* and *Adrb3* mRNAs as well as mRNA for the β-adrenergic target gene *Hsl* (***Figure 6A***). Activin B had no effect on adipocytes derived from *Alk7* knock-out MEFs (***Figure 6B***), indicating that its effects on adipocyte gene expression were mediated by ALK7. Activin B also downregulated expression of mRNA encoding PPARγ, a master regulator of adipogenesis (***Figure 6A***). However, the effects of activin B on adipocyte *Adrb* mRNA expression are likely to be independent of PPARγ, since inhibition of PPARγ had no significant effect on *Adrb3* mRNA levels and activin B could suppress *Adrb3* mRNA expression even in the presence of the PPARγ inhibitor or the PPARγ agonist rosiglitazone (***Figure 6—figure supplement 1***). Conversely, the related ligand activin A, which does not stimulate ALK7 signaling, enhanced adipocyte

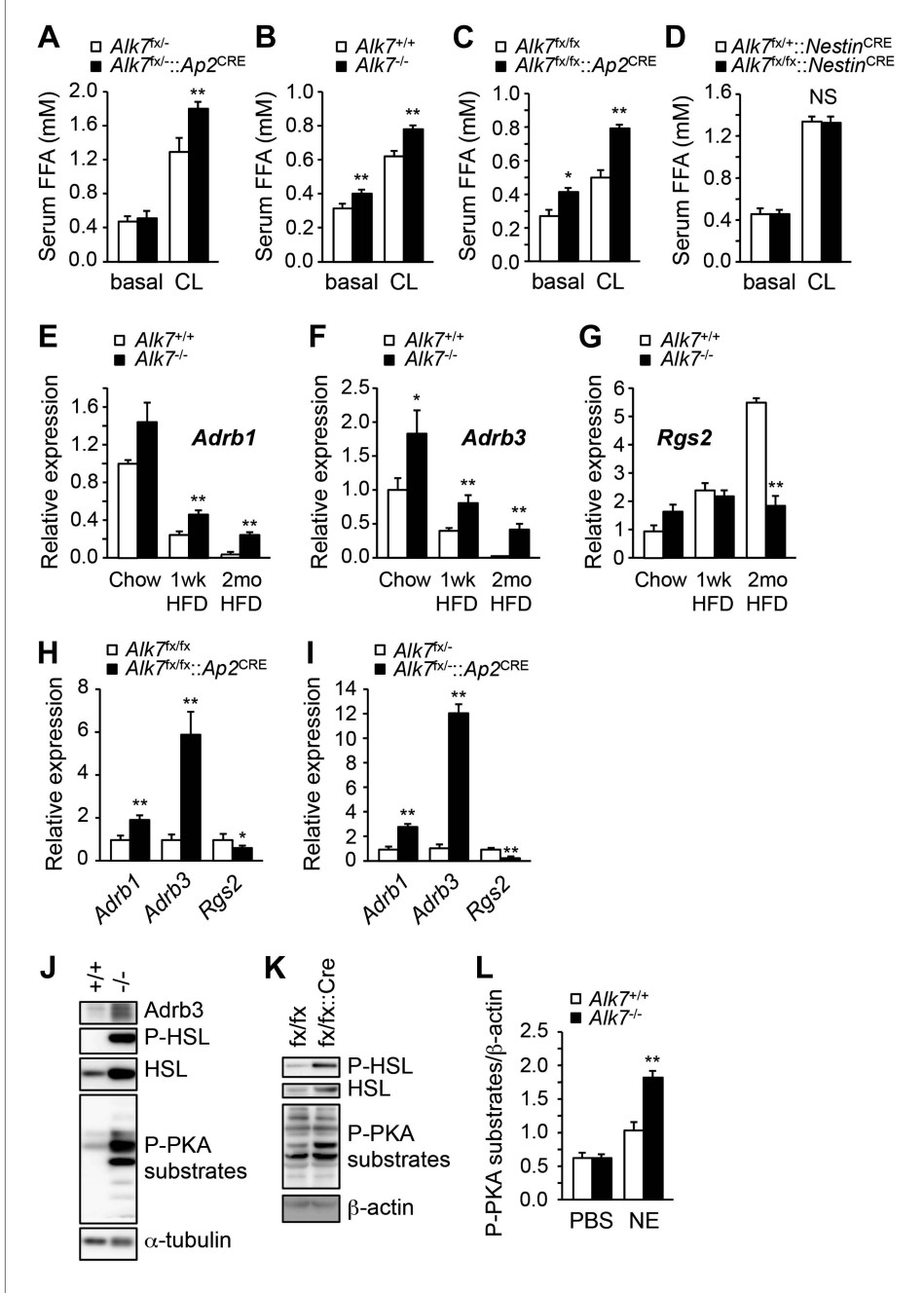

**Figure 5**. Enhanced catecholamine sensitivity and β-adrenergic signaling in adipose tissue of global and fat-specific *Alk7* knock-out mice on a high-fat diet. (**A**) Basal and CL316243 (CL)-stimulated serum free fatty acids (FFA) in fat-specific *Alk7*^fx/−^::*Ap2*^CRE^ knock-out mice on a chow diet. N = 6 mice per group. (**B–D**) Basal and CL316243-stimulated lipolysis in global *Alk7*^−/−^ (**B**), fat-specific *Alk7*^fx/fx^::*Ap2*^CRE^ (**C**), and brain-specific *Alk7*^fx/fx^::*Nestin*^CRE^ (**D**) knock-out mice after 16 weeks on a high fat diet (HFD). N = 6 mice per group. (**E–G**) Relative mRNA expression of *Adrb1* (**E**), *Adrb3* (**F**), and *Rgs2* (**G**) assessed by Q-PCR in epididymal adipose tissue of global *Alk7*^−/−^ knock-out mice on chow and after 1 week (wk) or 2 months (mo) on HFD. N = 6 mice per group. (**H** and **I**) Relative mRNA expression of *Adrb1*, *Adrb3*, and *Rgs2* assessed by Q-PCR in epididymal adipose tissue of fat-specific *Alk7*^fx/fx^::*Ap2*^CRE^ (**H**) and *Alk7*^fx/−^::*Ap2*^CRE^ (**I**) knock-out mice after 16 weeks on HFD. N = 6 mice per group. (**J** and **K**) Levels of Adrb3, phospho-HSL, total HSL, and phosphorylated PKA substrates assessed by Western blotting in epididymal adipose tissue of global *Alk7*^−/−^ (**J**) and fat-specific *Alk7*^fx/fx^::*Ap2*^CRE^ (**K**) knock-out mice after 16 weeks on HFD. The results shown are representative of four biological replicates. (**L**) Levels of phosphorylated PKA substrates were assessed by Western

*Figure 5. Continued on next page*

*Figure 5. Continued*

blotting in epididymal adipose tissue from 2-month-old global *Alk7*[−/−] knock-out mice on a chow diet following injection of norepinephrine (NE) or vehicle (PBS). Results show the levels of phosphorylated PKA substrates quantified by image analysis after normalization to β-actin. N = 3 mice per group. *p < 0.05; **p < 0.01; NS, non-significant (mutant vs control). All error bars show mean ± SEM.

The following figure supplement is available for figure 5:

**Figure supplement 1**. Epinephrine and norepinephrine levels in adipose tissue of *Alk7* knock-out mice after 16 weeks on a high fat diet.

PPARγ mRNA expression but had no significant effect on *Adrb2*, *Adrb3*, and *Hsl* mRNA levels in wild type adipocytes (*Figure 6C*). Expression of *Adrb2*, *Adrb3*, and *Hsl* mRNAs was also significantly reduced in *Alk7* knock-out adipocytes following ALK7 adenovirus-mediated overexpression (*Figure 6D*), which leads to ligand-independent receptor activation, showing that ALK7 is sufficient to suppress β-AR expression and signaling in adipocytes. In addition, activin B antagonized the stimulatory effects of

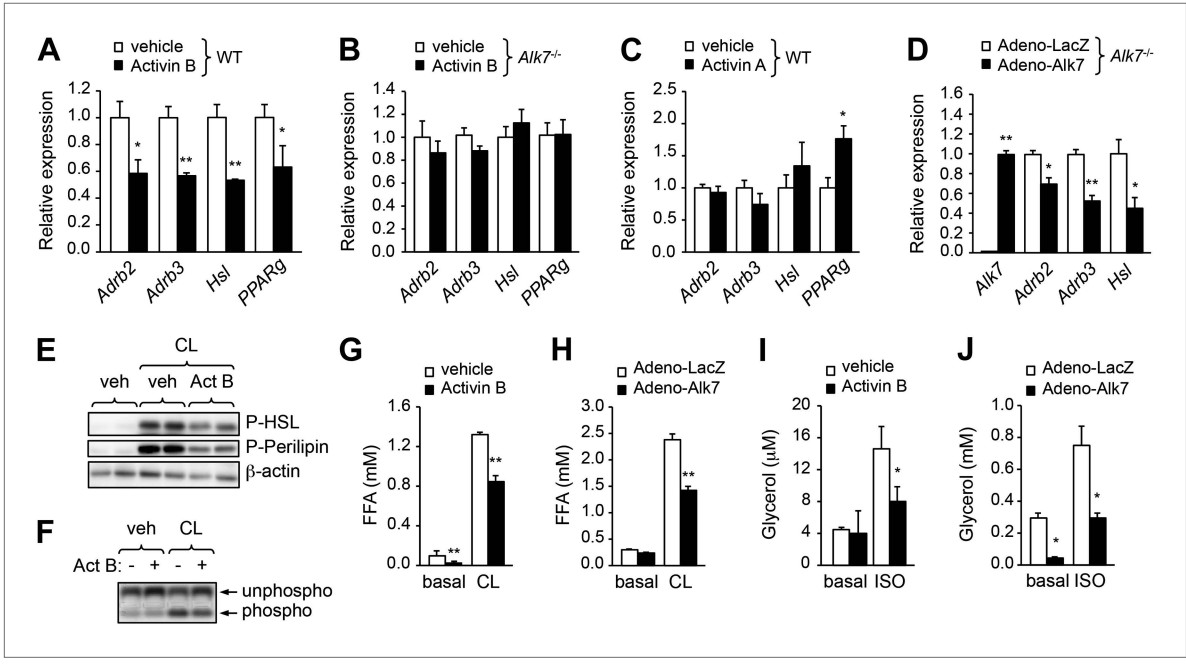

**Figure 6**. ALK7 signaling negatively regulates catecholamine sensitivity and β-adrenergic signaling in mouse and human adipocytes. (**A**) Relative mRNA expression of *Adrb2*, *Adrb3*, *Hsl*, and *PPARg* assessed by Q-PCR in adipocytes derived from wild type (WT) mouse embryonic fibroblasts (MEFs) following stimulation with activin B. N = 4 wells per condition. (**B**) Relative mRNA expression of *Adrb2*, *Adbr3*, *PPARg*, and *Hsl* assessed by Q-PCR in adipocytes derived from *Alk7* knock-out MEFs after stimulation with activin B. N = 4 wells per condition. (**C**) Relative mRNA expression of *Adrb2*, *Adbr3*, *PPARg*, and *Hsl* assessed by Q-PCR in adipocytes derived from WT MEFs following stimulation with activin A. N = 4 wells per condition. (**D**) Relative mRNA expression of *Alk7*, *Adrb2*, *Adrb3*, and *Hsl* assessed by Q-PCR in adipocytes derived from WT MEFs following adenovirus-mediated ALK7 overexpression. N = 4 wells per condition. (**E**) Levels of phospho-HSL and phospho-perilipin assessed by Western blotting in MEF-derived adipocytes after stimulation with β3-AR-specific agonist CL316243 (CL) or vehicle (veh) in the presence or absence of activin B (Act B). Independent biological duplicates are shown. (**F**) Assay of PKA activity in lysates of MEF-derived adipocytes after stimulation with β3-AR-specific agonist CL316243 (CL) or vehicle (veh) in the presence or absence of activin B (Act B). Independent biological duplicates are shown. (**G** and **H**) Basal and CL316243-stimulated lipolysis in MEF-derived adipocytes following stimulation with activin B (**G**) or adenovirus-mediated ALK7 overexpression (**H**). FFA: free fatty acids. N = 4 wells per condition. (**I** and **J**) Basal and isoproterenol (ISO)-stimulated lipolysis in human adipocytes following stimulation with activin B (**I**) or adenovirus-mediated ALK7 overexpression (**J**). N = 4 wells per condition. *p < 0.05; **p < 0.01; NS, non-significant (activin treatment vs vehicle or Adeno-Alk7 vs Adeno-LacZ, as indicated). All error bars show mean ± SEM.

The following figure supplement is available for figure 6:

**Figure supplement 1**. The effects of activin B on adipocyte *Adrb* mRNA expression are independent of PPARγ.

the β₃-AR agonist on the phosphorylation of HSL, perilipin, and PKA substrates in wild type adipocytes (*Figure 6E,F*). In line with these findings, activation of ALK7 signaling by either ligand or adenovirus-mediated overexpression suppressed β-agonist induced lipolysis in wild type adipocytes (*Figure 6G,H*). Interestingly, both activin B and adenovirus-mediated ALK7 overexpression also suppressed β-agonist stimulated lipolysis in human adipocytes (*Figure 6I,J*). Together, these results indicate that acute activation of ALK7 is sufficient to suppress β-adrenergic signaling in both mouse and human adipocytes.

### Acute inhibition of ALK7 signaling in adult mice through a chemical-genetic approach prevents diet-induced catecholamine resistance and ameliorates obesity

Finally, we sought to determine whether acute blockade of ALK7 signaling in adult mice, by-passing possible developmental effects, could also enhance catecholamine sensitivity in adipose tissue and ameliorate diet-induced obesity. To this end, we devised a chemical-genetic approach to acutely inhibit the ALK7 kinase in adult mice using synthetic ATP analogues (*Bishop et al., 1998*, *2001*). An analogue-sensitive kinase allele of *Alk7* (termed *Alk7*ASKA) was engineered by mutation of two residues in the active site of the ALK7 kinase. Substitution of the 'gatekeeper' residue Ser²⁷⁰ with Gly creates an extra pocket in the ALK7 active site, which can be further expanded by mutating the adjacent residue Leu²⁵⁰ to Val. As assessed in transfected cells, the mutations had no effect per se on ligand-mediated signaling, but rendered the mutant receptor sensitive to inhibition by ATP competitive inhibitors, such as 2NaPP1 (*Figure 7—figure supplement 1*). 2NaPP1 treatment did not affect signaling by the wild type ALK7 receptor (*Figure 7—figure supplement 1*). Knock-in mice were generated carrying the two mutations in the *Alk7* locus (*Figure 7—figure supplement 2*). *Alk7*ASKA mice developed normally, and displayed normal growth and normal fasting insulinemia and glycemia (data not shown). Activin B signaling was suppressed by 1NaPP1 (an isomer of 2NaPP1 with established in vivo stability [*Savitt et al., 2012*]) in adipocytes derived from *Alk7*ASKA homozygote mice but not from wild type controls (*Figure 7A,B*). Residual activin B activity in *Alk7*ASKA adipocytes treated with 1NaPP1 was likely due to expression of the related receptor ALK4, which can also respond to activin B by activating Smad3. We subjected cohorts of 2-month-old *Alk7*ASKA homozygote and wild type mice to a high fat diet together with twice daily injection of 30 mg/kg 1NaPP1 or vehicle, and monitored their weight daily during a period of 2 weeks. *Alk7*ASKA and wild type mice gained weight at the same rate on a high fat diet, increasing by approximately 4 g by the end of the second week (*Figure 7C,D*). There was no weight gain on a chow diet for either genotype (*Figure 7C,D*). Injection of 1NaPP1 significantly reduced the weight gain rate in *Alk7*ASKA mice, which on average increased by only 1.6 g after 2 weeks on a high fat diet (*Figure 7C*). 1NaPP1 had no effect on wild type mice (*Figure 7D*). These results indicate that acute disruption of ALK7 signaling can protect adult mice from diet-induced obesity. Analogue treatment significantly reduced fat accumulation in epididymal and retroperitoneal adipose tissue depots in *Alk7*ASKA but not wild type mice (*Figure 7E,F*). Adipocyte cell size was also significantly reduced in *Alk7*ASKA mice treated with 1NaPP1 compared to vehicle (*Figure 7G,H*). We also observed enhanced induction of lipolysis by the β₃-AR-specific agonist CL316243 in adipose tissue biopsies extracted from *Alk7*ASKA mice that had been treated with 1NaPP1 (*Figure 8A*). Moreover, treatment with 1NaPP1 increased the levels of *Adrb3* mRNA in epididymal and retroperitoneal adipose tissue of *Alk7*ASKA mice on a high fat diet (*Figure 8B*). In line with enhanced catecholamine sensitivity, 1NaPP1 treatment attenuated the suppressive effects of a high fat diet on adrenergic signaling in adipose tissue of *Alk7*ASKA mice, as assessed by the levels of phosphorylated HSL and PKA substrates (*Figure 8C–E*). Together, these data indicated that acute disruption of ALK7 signaling in adult mice can uncouple nutrient overload from catecholamine resistance in adipose tissue, resulting in reduced fat accumulation and decreased weight gain on a high fat diet.

## Discussion

In mice and humans, nutrient overload induces catecholamine resistance in adipose tissue, thereby suppressing lipolysis and fatty acid oxidation, and enhancing fat accumulation. This is an efficient adaptive mechanism for energy storage during times of abundant food supply, enhancing survival upon starvation. In the modern industrialized world, however, the effortless availability of foods with high caloric content, together with reduced physical activity, has resulted in an obesity pandemic. Remarkably, the mechanisms linking nutrient overload to catecholamine resistance in adipose tissue are not well understood. Clarifying these mechanisms is important, as selective enhancement of

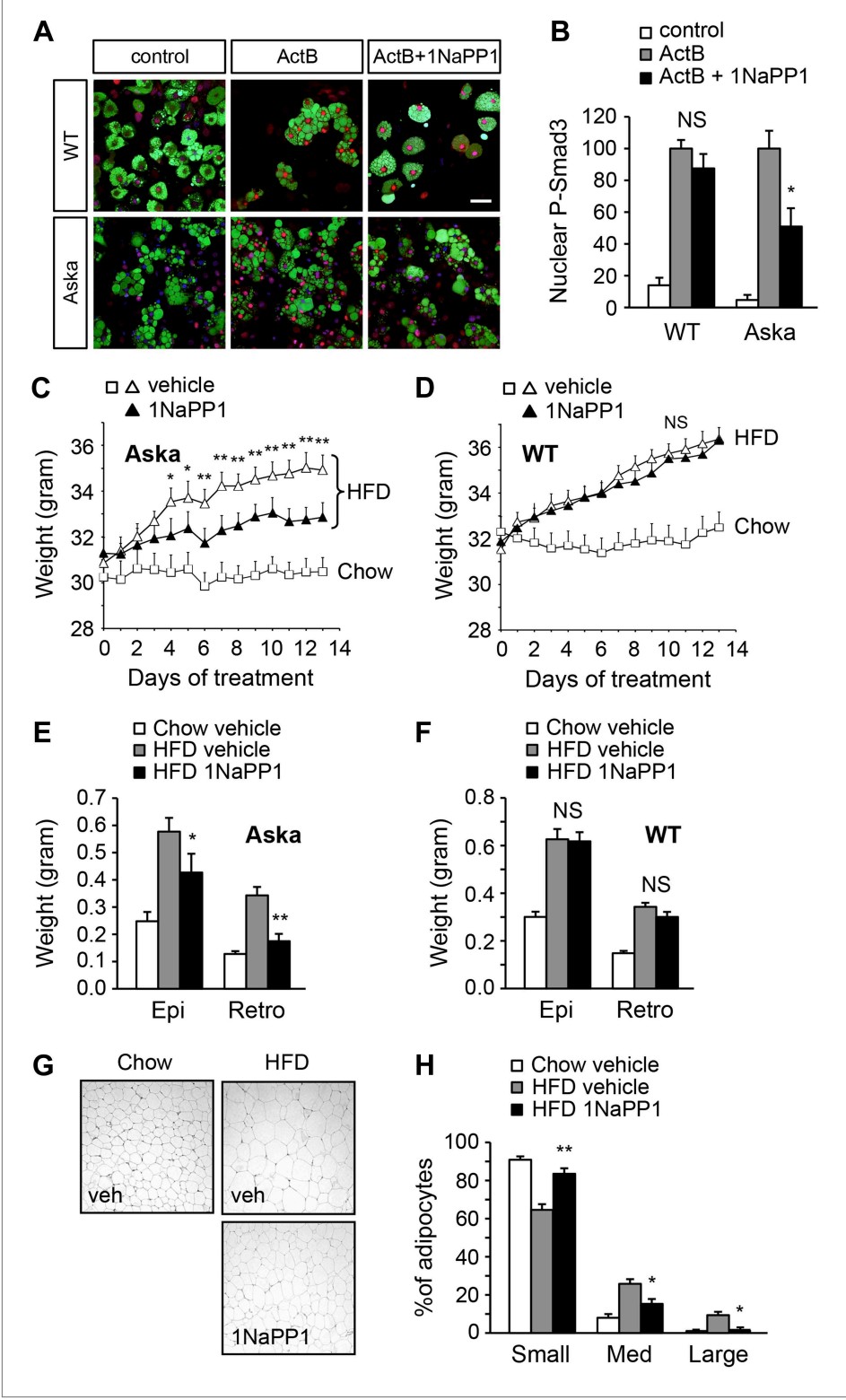

**Figure 7**. Acute inhibition of ALK7 signaling in adult mice through a chemical-genetic approach reduces diet-induced weight gain and fat accumulation. (**A** and **B**) Activin B signaling was assessed by p-Smad3 nuclear translocation in mouse embryonic fibroblast (MEF)-derived adipocytes from wild type (WT) or $Alk7^{ASKA}$ (Aska) mice in the presence and absence of ATP competitive inhibitor 1NaPP1. Nuclear p-Smad3 (red) was specifically evaluated by
*Figure 7. Continued on next page*

*Figure 7. Continued*

immunohistochemistry in adipocytes, identified by BODIPY 493/503 staining (green). Representative photomicrographs are shown in (**A**). Scale bar, 50 μm. Quantitative analysis is shown in (**B**). N = 3 independent experiments each performed in triplicate. (**C** and **D**) Weight gain on chow (squares) and a high fat diet (HFD, triangles) in *Alk7*^ASKA (**C**) and wild type (**D**) mice treated with 1NaPP1 (solid triangles) or vehicle (open squares and triangles). N = 6 mice per group in (**C**), N = 9 in (**D**). (**E** and **F**) Weights of epididymal (Epi) and retroperitoneal (Retro) fat depots in *Alk7*^ASKA (**E**) and wild type (**F**) mice after chow (open bars) or 2 weeks on HFD treated with 1NaPP1 (black bars) or vehicle (gray bars). N = 6 mice per group in (**E**), N = 7 in (**F**). (**G** and **H**) Adipocyte cell size in *Alk7*^ASKA mice after chow or HFD as visualized by hematoxylin-eosin staining in tissue sections of epididymal adipose tissue of *Alk7*^ASKA mice after chow or 2 weeks on HFD treated with 1NaPP1 or vehicle (veh) (**G**). Quantitative analysis is shown in (**H**). Small, 400–5000 μm$^2$; Med, 5000–10,000 μm$^2$; Large, 10,000–20,000 μm$^2$. N = 4 mice per group (four sections per mouse). *$p < 0.05$; **$p < 0.01$; NS, non-significant (1NaPP1 vs vehicle). All error bars show mean ± SEM.

The following figure supplements are available for figure 7:

**Figure supplement 1**. Validation of *Alk7*^ASKA allele in transfected R4-2 cells.

**Figure supplement 2**. A chemical-genetic approach for acute inactivation of ALK7 in adult mice.

catecholamine sensitivity and β-adrenergic signaling in adipose tissue could be useful in the treatment of obesity. Obesity is thought to induce catecholamine resistance in adipose tissue by downregulating β-AR expression and interfering with downstream β-adrenergic signaling pathways, such as PKA activity (*Reynisdottir et al., 1994*; *Arner, 1999*; *Jocken et al., 2008*). Our results demonstrate that ALK7 signaling contributes to catecholamine resistance by limiting both β-AR expression and signaling in adipocytes during a high fat diet. Signaling pathways are known to be regulated by negative feedback loops that limit intracellular responses upon sustained stimulation. Adrenergic signaling in adipose tissue self-attenuates under a high fat diet through PKA-mediated upregulation of Rgs2 (*Song et al., 2010*). Interestingly, a mutation in the *Rgs2* promoter that increases Rgs2 expression has been shown to enhance susceptibility to metabolic syndrome in humans (*Freson et al., 2007*). However, it is unclear how elevated levels of Rgs2 can be sustained in adipose tissue during persistent catecholamine resistance. In our studies, disruption of ALK7 in adipose tissue largely prevented the upregulation of Rgs2 under a high fat diet, suggesting that ALK7 signaling may contribute to the maintenance of diet-induced Rgs2 expression, possibly by cooperating with intermediate regulators, such as Crtc3 (*Song et al., 2010*).

Our studies in fat-specific *Alk7* knock-out mice and *Alk7*^ASKA knock-in mice demonstrate that ALK7 functions cell-autonomously and homeostatically in adult adipocytes to regulate catecholamine sensitivity in response to the diet. Adipocytes lacking ALK7 retain significant levels of β-AR expression and adrenergic signaling, allowing sustained lipid catabolism and increased energy expenditure. Although such a short-circuit would clearly be maladaptive upon starvation, it offers a therapeutic opportunity for the treatment of obesity. The fact that activin B and adenovirus-mediated ALK7 overexpression suppressed β-agonist stimulated lipolysis in human adipocytes suggests functional conservation of the ALK7 signaling pathway in human diet-induced obesity. A high fat diet and obesity increase adipocyte expression of the two main ALK7 ligands activin B and GDF-3 (*Witthuhn and Bernlohr, 2001*; *Sjöholm et al., 2006*; *Hoggard et al., 2009*), suggesting enhanced ALK7 signaling in obesity. In line with this, diet-induced obesity elevates the level of activated phospho-Smad3 in mouse adipose tissue (*Yadav et al., 2011*). Mouse and human *Alk7* mRNA expression is highest in adipose tissue, but undetectable in heart, liver or muscle (*Kang and Reddi, 1996*; *Rydén et al., 1996*; *Carlsson et al., 2009*; *Murakami et al., 2012*). Thus, modulation of ALK7 signaling may offer an alternative approach to regulate catecholamine sensitivity and β-adrenergic signaling specifically and selectively in adipose tissue. We propose that ALK7 represents a novel link between nutrient overload and catecholamine resistance in adipose tissue, and suggest that strategies to suppress ALK7 signaling may be beneficial to combat human obesity.

## Materials and methods

### Animals

Mice were housed under a 12 hr light–dark cycle, and fed a standard chow diet or a high fat diet (HFD, 60% of calorie from fat; ResearchDiet). The following transgenic mouse lines were used for

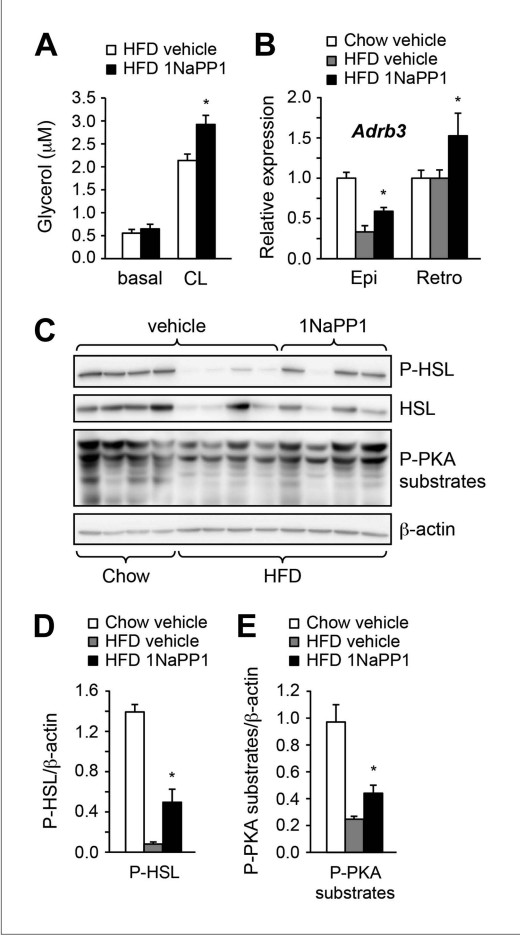

**Figure 8**. Acute inhibition of ALK7 signaling in adult mice reduces diet-induced catecholamine resistance. (**A**) Basal and CL316243 (CL)-stimulated lipolysis in adipose tissue biopsies extracted from *Alk7*[ASKA] mice after 2 weeks on a high fat diet (HFD) treated with 1NaPP1 (solid bars) or vehicle (open bars). N = 6 mice per group. Epi, epididymal; Retro, retroperitoneal. (**B**) Relative mRNA expression levels of *Adrb3* assessed by Q-PCR in epididymal (Epi) adipose tissue of *Alk7*[ASKA] mice after chow (open bars) or 2 weeks on HFD treated with 1NaPP1 (black bars) or vehicle (gray bars). N = 6 mice per group. (**C–E**) Levels of phospho-HSL (p-HSL) and phosphorylated PKA (P-PKA) substrates assessed by Western blotting in epididymal adipose tissue of *Alk7*[ASKA] mice after chow or 2 weeks on HFD treated with 1NaPP1 or vehicle. Histograms show quantification by image analysis of P-HSL levels (**D**) and levels of phosphorylated PKA substrates (**E**) normalized to β-actin levels. N = 4 mice per group. *$p < 0.05$; **$p < 0.01$; NS, non-significant (1NaPP1 vs vehicle). All error bars show mean ± SEM.

experiments: *Alk7* knock-out (*Jörnvall et al., 2004*), *Ap2*[CRE] (*He et al., 2003*), *Nestin*[CRE] (*Tronche et al., 1999*), *Alk7*[fx] conditional knock-out (this study), and *Alk7*[ASKA] knock-in (this study). The *Alk7*[fx] conditional knock-out allele was generated by inserting *loxP* sites flanking exons 5 and 6 (*Figure 1—figure supplement 1*). Fat-specific knock-out lines were generated by crossing *Alk7*[fx] or *Alk7*[fx/] to *Ap2*[CRE]. Brain-specific knock-out lines were generated by crossing *Alk7*[fx] to *Nestin*[CRE]. The *Alk7*[ASKA] knock-in allele was generated by introducing L250V and S270G mutations into the *Alk7* gene exons 4 and 5, respectively (*Figure 7—figure supplement 2*). Targeting vectors were generated using BAC clones from the C57BL/6J RPCIB-731 BAC library and transfected into the TaconicArtemis C57BL/6N Tac ES cell line. Gene-targeted mice were generated at TaconicArtemis (Germany) by standard methods. Animal experiments were approved by Stockholm North Ethical Committee for Animal Research.

## Body composition and whole-body energy homeostasis

Fat and lean mass were measured using a body composition analyzer (EchoMRI Medical System, Houston, TX, USA). Indirect calorimetry, food intake, and locomotor activity were determined using a comprehensive laboratory animal monitoring system (Columbus Instruments, Columbus, OH, USA) as previously described (*Chibalin et al., 2008*). Mice were housed individually with ad libitum access to a high fat diet and water. Mice were acclimatized to the metabolic cages for 24 hr prior to a 24 hr period of automated recordings. Oxygen consumption ($VO_2$) and $CO_2$ production were determined by an open-circuit Oxymax. Accumulated $VO_2$ was presented in l/kg/day and energy expenditure was reported as heat production divided by kilograms of body weight (kcal/kg/day). Ambulatory locomotor activity (XAMB) was measured by consecutive beam breaks in adjacent beams during a 24 hr period and presented as counts/min.

## In vivo administration of 1NAPP1

*Alk7*[ASKA] and wild type mice (8 weeks old) kept on a chow diet were individually caged for 1 week, then switched to HFD or kept on a chow diet. 1NAPP1 was freshly formulated in PEG400 at a concentration of 12.5 mg/ml before use. Twice daily (9:30 AM and 5:30 PM, respectively) mice on HFD received subcutaneous injections of vehicle or 1NAPP1 (50 mg/kg/day).

## Glucose and insulin tolerance tests

Glucose tolerance tests were performed on animals kept on chow or a high fat diet after overnight fasting. Blood glucose was measured by tail tip bleeding using a glucometer (Accutrend; Roche, Sweden) at the indicated time points before and after intraperitoneal injection of 2 g/kg (body weight)

glucose. Serum insulin was measured from tail blood with an ultrasensitive mouse insulin ELISA kit (Mercodia, Sweden). For the insulin tolerance test, mice were fasted for 3 hr, then injected intraperitoneally with 1.5 IU/kg recombinant human insulin (Humulin R; Lilly, Sweden). Glucose levels were determined at the indicated time points as above.

## Histology

Fresh tissue samples were fixed in 10% formalin, dehydrated, embedded in paraffin, sectioned, and stained with hematoxylin and eosin. Photographs of hematoxylin and eosin stained cross-section slides were taken with a light microscope. Areas of individual adipocytes were measured with ImageJ software (National Institutes of Health) and used for quantification of cell size.

## Lipolysis assay

For in vivo lipolysis, mouse tail blood was taken from non-fasted mice for measurement of serum free fatty acids before and after intraperitoneal injection of the $\beta_3$-AR-specific agonist CL316243 (Sigma-Aldrich, Sweden) (100 µg/kg body weight) at the indicated times (20 min or 40 min). For ex vivo lipolysis, epididymal fat pieces weighing about 20 mg were incubated in Krebs–Ringer Bicarbonate Buffer (KRBH) containing 1% fatty acid-free BSA (Sigma-Aldrich) and glucose (2.5 mM). The samples were treated with either vehicle or CL316243 (1 µM) for 2 hr at 37°C with mild shaking at 150 rpm. After incubation, glycerol release was measured using a free glycerol reagent (Sigma-Aldrich), and normalized to the total amount of protein in adipose tissue samples. For in vitro lipolysis in MEF-derived or human adipocytes, differentiated adipocytes were washed with KRBH and treated with either vehicle, CL316243 (1 µM), Norepinephrine (1 µM), or isoproterenol (1 µM) for 3 hr, and glycerol release was measured as before.

## Lipid oxidation

Primary adipocytes were isolated from epididymal adipose tissue after collagenase II (1 mg/ml, Sigma-Aldrich) digestion, and were incubated with KRBH containing 1% fatty acid-free BSA (Sigma-Aldrich), glucose (2.5 mM), palmitic acid (100 µM), and $^3$H palmitic acid (0.5 µCi/ml). For normalization of counts by total protein, an aliquot of adipocytes was snap frozen without BSA and $^3$H palmitic acid incubation. After incubation for 5 hr at 37°C with shaking at 150 rpm, an equal volume of chloroform was added. After chloroform separation, 100 µl of the aqueous phase was further mixed with 900 µl 10% activated charcoal slurry thoroughly by shaking for 30 min. The mixture was subsequently centrifuged at maximum speed at room temperature for 10 min, and 100–200 µl supernatant was taken for assessment of $^3$H counts.

## Adipocyte differentiation and treatment

MEFs were prepared from 13.5-day wild type and *Alk7* mutant embryos, and differentiated into adipocytes according to a standard protocol (*Zhang et al., 2009*). Briefly, MEF cells were plated at about 50–70% confluence on 0.1% gelatin-coated dishes, cultured in DMEM, supplemented with 10% fetal bovine serum, 1 mM sodium pyruvate, MEM Non-Essential Amino Acids Solution (Invitrogen, Sweden), 0.5 mM 2-mercaptoethanol, and 100 U/ml penicillin and streptomycin (MEF medium). MEF monolayers were allowed to grow to confluence before initiation of adipogenesis. Adipogenesis was induced by incubation in MEF medium supplemented with 10 µg/ml insulin, 0.5 mM IBMX, 1 µM dexamethasone, and 0.5 µM rosiglitazone (adipocyte differentiation medium). After 2 days, the medium was changed to MEF medium supplemented with only 10 µg/ml insulin and 0.5 µM rosiglitazone (adipocyte maintenance medium), and then changed again every 2 days until full differentiation (i.e., day 8–10). Human preadipocytes were purchased from Lonza Biologics and differentiated into adipocytes according to the manufacturer's instructions. For activin B or activin A treatment, cells were incubated with 100 ng/ml activin B or activin A (R&D Systems, UK) in adipocyte maintenance medium for 48 hr before lipolysis assay or, alternatively, harvested for extraction of RNA and proteins. PPARγ agonist rosiglitazone was used at 0.5 µM; PPARγ antagonist T0070907 at 1 µM. Adenovirus particles were produced and amplified in HEK293 cells as previously described (*Fujii et al., 1999*). Adipocytes were infected with Adeno-ALK7 or Adeno-LacZ at 2.0 multiplicity of infection (MOI) for 48 hr before lipolysis assay or extraction of RNA. For assessment of nuclear p-Smad3, MEF-derived adipocytes were incubated for 150 min with 2 µM 1NaPP1 or vehicle (DMSO) then treated for 40 min with 20 ng/ml activin B, or left untreated, and fixed with 4% paraformaldehyde. Adipocytes were identified by BODIPY 493/503 staining and p-Smad3 was assessed by immunocytochemistry with antibodies from Epitomics. DAPI was used for nuclear staining. Images were obtained with a Zeiss laser confocal microscope.

The intensity of nuclear p-Smad3 was determined using Zen software (Zeiss, Germany) in adipocytes from three random fields in three different wells per condition. The number of cells displaying nuclear p-Smad3 intensity above an arbitrary threshold was determined in each field, added up for each well, and averaged across the three wells of each condition. All values were then normalized to the number of responding cells in cultures of wild type adipocytes stimulated with activin B (set to 100). The experiment was repeated three times and the results were averaged across the three experiments.

## Biochemical assays

Serum triglyceride (Infinity triglyceride reagent; Thermo DMA), free fatty acids (Free Fatty Acids Half Micro Test; Roche), leptin (mouse leptin ELISA; Abcam, UK), insulin (Ultrasensitive mouse insulin ELISA; Mercodia), and tissue ATP contents (ATPLite; PerkinElmer) were measured with commercial kits according to the manufacturer's instructions. For quantification of mitochondria DNA, total adipose tissue DNA was extracted with a DNeasy kit (Qiagen, Sweden) and 1 ng was used for Q-PCR quantification of the copy number for mitochondrial encoded gene *cytochrome B*. This was normalized to the copy number of the nuclear gene H19. Citrate synthase activity was measured as described (*Srere et al., 1963*) in tissue samples homogenized in RIPA buffer. Epinephrine and norepinephrine were measured in adipose tissue extracts by a commercial ELISA kit according to the manufacturer's instructions (Labor Diagnostika Nord, Sweden). A PKA activity assay was performed with PepTag Non-Radioactive cAMP-Dependent Protein Kinase Assay (Promega, Sweden).

## Tissue isolation, RNA preparation, and quantitative PCR (Q-PCR)

For isolation of adipocyte and stromal-vascular fractions (SVF), tissues were dissected from epididymal depots of 8-week-old mice, minced with forceps in PBS, and washed three times with 0.1% BSA KRBH. Digestion was carried out for 90–120 min in 2% BSA KRBH containing 0.2 mg/ml collagenase II (Sigma-Aldrich) with constant agitation. Digested tissue was filtered through 250 µm nylon mesh and centrifuged for 10 min at 1000 rpm ($200 \times g$) to separate floating adipocytes. The supernatant and the pellet were further centrifuged for 10 min at 1500 rpm ($240 \times g$). The resulting pellet (SVF) and the adipocytes were resuspended in lysis buffer and kept at −80°C. 'Browning' of subcutaneous adipose tissue was induced by chronic treatment with CL316243. A number of 12–16-week-old mice were injected daily intraperitoneally with either 1 mg/kg CL316243 (Sigma) or PBS only (vehicle) for 7 days. Inguinal subcutaneous adipose tissue was isolated from these mice. Total RNA from tissues or cells was isolated with an RNAeasy kit (Qiagen) according to the manufacturer's instructions. Isolated RNA samples were digested with DNase prior to reverse transcription with Superscript II (Invitrogen). Q-PCR of cDNA samples was performed on a ABI StepOne Plus instrument (Applied Biosystems, Sweden) using SYBR Green master mix (Applied Biosystems) and primers as indicated in *Supplementary file 1*.

## Cell transfection and luciferase assay

The R4-2 cell line is a derivative of MvLu1 mink lung epithelial cells that expresses low levels of type 1 TGF-β receptors (*Andersson et al., 2008*). R4-2 cells were cultured in 24-well plates in DMEM medium supplemented with 10% serum and antibiotics. They were transfected using Lipofectamine 2000 (Invitrogen) with expression plasmids carrying wild type or Aska variants of the rat Alk7 cDNA, the Smad3-dependent luciferase reporter plasmid CAGA-Luc (*Dennler et al., 1998*), and a Renilla plasmid for normalization. At 24 hr after transfection, cells were stimulated with 25 ng/ml activin B together with 1 µM 2NaPP1 (solubilized in DMSO) or DMSO only (vehicle). Then 24 hr after stimulation, luciferase activity was analyzed using the Dual-Luciferase Reporter Assay System (Promega) in a 1450 Microbeta Jet counter (Wallac). For assays of p-Smad2 by Western blotting, cells were treated 48 hr after transfection with 1 µM 2NaPP1 for 3 hr prior to addition of 25 ng/ml activin B for an additional hour. Monolayers were then processed for Western blotting as described below.

## Western blot analysis

Protein extraction and Western blotting were performed according to standard protocols. Briefly, snap-frozen adipose tissue or cell samples were homogenized in ice-cold RIPA buffer (50 mM Tris–HCl pH 7.4, 1% NP-40, 0.25% sodium deoxycholate, 150 mM sodium chloride, 1 mM EDTA) containing 1 mM sodium orthovanadate, 1 mM sodium fluoride, and proteinase inhibitor cocktail (Roche), and centrifuged at $13,000 \times g$ for 15 min to collect supernatants. Supernatants (30 µg protein) were used for reducing SDS PAGE and Western blotting. Primary antibodies against p-HSL, HSL, p-PKA substrates,

p-Perilipin, p-Smad2, β-actin, and α-tubulin (Cell Signaling, Sweden) were used at 1:2000 dilution. The levels of target proteins were quantified by the intensity of Western blot bands using ImageJ software (National Institutes of Health).

## Statistical analysis

Statistical significance was determined by using unpaired two-tailed or one-tailed Student's *t* tests and one-way ANOVA. Differences were considered significant at a p value less than 0.05. Quantified data are presented as mean ± SEM.

## Acknowledgements

We thank Peter Arner, Mikael Ryden, Juleen Zierath, and Anna Krook for discussion; Annica Ho, Olov Andersson, and Philippe Bertolino for preliminary Aska experiments; Peter ten Dijke for reagents; and Annika Andersson and Berit Engst for technical assistance. This work was supported by grants from the European Research Council, Swedish Research Council, Strategic Research Program in Diabetes of Karolinska Institutet, Swedish Cancer Society, Knut and Alice Wallenberg Foundation (Wallenberg Scholars Program), and the National University of Singapore.

## Additional information

### Funding

| Funder | Grant reference number | Author |
| --- | --- | --- |
| European Research Council | | Carlos F Ibanez |
| Vetenskapsrådet | | Carlos F Ibanez |
| Cancerfonden | | Carlos F Ibanez |
| Strategic Programme in Diabetes KI | | Carlos F Ibanez |
| Knut och Alice Wallenbergs Stiftelse | | Carlos F Ibanez |
| National University of Singapore | | Carlos F Ibanez |
| National Medical Research Council of Singapore | CIRG13nov037 | Carlos F Ibanez |

The funders had no role in study design, data collection and interpretation, or the decision to submit the work for publication.

### Author contributions

TG, Conception and design, Acquisition of data, Analysis and interpretation of data, Drafting or revising the article; PM, AM, MB, Acquisition of data, Analysis and interpretation of data; CZ, KMS, Conception and design, Analysis and interpretation of data; CFI, Conception and design, Analysis and interpretation of data, Drafting or revising the article

### Ethics

Animal experimentation: Animal experiments were approved by the Stockholm's North Ethical Committee for Animal Research as per ethical protocols N111/14, N115/14 and N36/14.

## Additional files

### Supplementary file

• Supplementary file 1. Primers for qPCR. Sequences of primers used for qPCR are given.

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
