## [Decision Letter]

Thank you for sending your work entitled "Adipocyte ALK7 links nutrient overload to catecholamine resistance in obesity" for consideration at *eLife.* Your article has been favorably evaluated by Randy Schekman (Senior editor) and 3 reviewers, two of whom are members of our Board of Reviewing Editors.

The Reviewing editor and the other reviewers discussed their comments before we reached this decision, and the Reviewing editor has assembled the following comments to help you prepare a revised submission.

The reviewers were in agreement that the work was exciting and potentially of interest to the readers of *eLife*. However, several major issues were identified that need to be addressed in order to strengthen the conclusions drawn. In particular, all three reviewers raised concerns about the animal models and identified a need for additional metabolic characterization and analysis of thermogenesis.

Major concerns:

1) Mouse models. The authors cannot definitively conclude that the effects are specific to adipose tissue with the models used. First, the Fabp4 promoter also drives expression in non-adipocyte cells, including myeloid cells. Second, the authors make use of a whole body heterozygote knockout mouse for their tissue-specific loss-of-function studies. In their previous work, the authors demonstrated that ALK7 ablation causes beta cell dysfunction and metabolic disease. To address these concerns, the authors should examine deletion of ALK7 in macrophages and brain and address whether there is an intrinsic effect of ALK7 deletion on macrophage M1/M2 polarization. The authors should also soften the conclusions that the effects are adipocyte intrinsic, as this statement cannot be adequately supported using the *Ap2cre* model.

2) Energy balance. The authors should address whether the ALK7 depletion or inhibition actually enhances browning of subcutaneous adipose tissue. Chronic administration of the CL compound or cold exposure yields increasing numbers of brite/beige adipocytes in normal mice. The authors should test whether "browning" is enhanced under such protocols. The data for expression of UCP1, Cidea and other brown fat markers should also be included for subcutaneous WAT and BAT.

3) Metabolic characterization. Previous work by this group demonstrated that *Alk7*^-/-^ mice have a metabolic dysfunction but the current manuscript does not include any data testing whether glucose metabolism or insulin sensitivity is perturbed in the new Alk7 mouse models. The authors should perform glucose tolerance tests, insulin tolerance tests, and fasting glucose and insulin measurements.

---

## [Author Response]

We would like to thank the reviewers for taking the time to read our manuscript and for their constructive comments.

The revised version of the manuscript includes 12 new panels with additional data as follows:

Absence of *Alk7* mRNA expression in adipose tissue macrophages is shown in panels A to D of Figure 1—figure supplement 2.

Normal glucose and insulin tolerance responses in fat-specific mutant mice is shown in panels E to H of revised Figure 2.

Absence of enhanced “browning” in subcutaneous adipose tissue of global and fat-specific knock-out mice is shown in panels A to D of Figure 3—figure supplement 1.

*1) Mouse models. The authors cannot definitively conclude that the effects are specific to adipose tissue with the models used. First, the Fabp4 promoter also drives expression in non-adipocyte cells, including myeloid cells*.

ALK7 is not expressed in myeloid cells of adipose tissue, so there is no problem with recombination in such cells. We have now included data (new Figure 1—figure supplement 2) demonstrating absence of Alk7 mRNA expression in stromal-vascular fraction (SVF) of adipose tissue, which contains adipose tissue macrophages, and in spleen. These data demonstrate that Alk7 mRNA in adipose tissue is clearly of adipocyte origin.

*Second, the authors make use of a whole body heterozygote knockout mouse for their tissue-specific loss-of-function studies*.

Most of the data presented is derived from replicate studies performed in both flox/flox and flox/– mice. This includes all the major phenotypes reported, such as weight gain in HFD, weight of fat depots, leptin levels, macrophage markers, basal and CL-stimulated lipolysis, *Adrb* and *Rgs2* adipocyte expression, and downstream noradrenergic signaling (i.e. HSL, PKA). Thus, flox/flox mice show all the major phenotypes described here, including resistance to diet-induced obesity, reduced diet-induced fat accumulation and adipose tissue inflammation, enhanced lipolysis, reduced diet-induced catecholamine resistance and enhanced noradrenergic signaling. In fact, flox/flox and flox/– mice are very similarly affected in all these phenotypes. We therefore do not think that the use of flox/– mice for some of the experiments has affected the main conclusions of our study. These mice are there simply to address a possible concern about residual Alk7 expression in flox/flox mice.

*In their previous work, the authors demonstrated that ALK7 ablation causes beta cell dysfunction and metabolic disease. To address these concerns, the authors should examine deletion of ALK7 in macrophages and brain and address whether there is an intrinsic effect of ALK7 deletion on macrophage M1/M2 polarization*.

Our manuscript presented an analysis of Alk7 deletion in brain using the Nestin-CRE driver. We showed that Nestin-CRE conditional Alk7 mutant mice lack

ALK7 expression in brain but not in adipose tissue, and are indistinguishable from wild type mice in diet-induced weight gain, fat deposition, insulin levels and basal as well as CL-induced lipolysis. We conclude from this that deletion of ALK7 in brain does not confer resistance to obesity, accumulation of fat tissue and diet-induced catecholamine resistance, all of which are improved in the Ap2- CRE model. Also, as indicated above, there cannot be an intrinsic effect of ALK7 in macrophage M1/M2 polarization because Alk7 is not expressed in macrophages. The changes that we see in the M1/M2 profile of mutant adipose tissue must be non-cell autonomous to macrophages. Our interpretation is that such changes are most likely the result of reduced fat accumulation and improved catecholamine responses in adipocytes

*The authors should also soften the conclusions that the effects are adipocyte intrinsic, as this statement cannot be adequately supported using the aP2 Cre model*.

Our view is that the analysis described above, together with the in vitro data shown in Figure 6, demonstrates that the effects we describe are cell-autonomous to adipocytes. First, deletion of ALK7 in brain does not affect obesity, fat deposition, lipolysis or catecholamine resistance. Second, *Alk7* is not expressed in myeloid cells of adipose tissue. Third, the in vitro experiments in MEF-derived adipocytes show that ALK7 is necessary (Figure 6) and sufficient (Figure 6) for regulation of *Adrb* and *Hsl* expression in isolated adipocytes, and sufficient to regulate lipolysis in both mouse (Figure 6) and human (Figure 6) adipocytes. We are confident that, together, these results allow us to establish several functions of the ALK7 receptor that are intrinsic (i.e. cell-autonomous) to adipocytes.

*2) Energy balance. The authors should address whether the ALK7 depletion or inhibition actually enhances browning of subcutaneous adipose tissue. The data for expression of UCP1, Cidea and other brown fat markers should also be included for subcutaneous WAT and BAT*.

Alk7 deletion does not enhance browning of subcutaneous adipose tissue. This is now shown in panels A and B of the new Figure 3—figure supplement 1. We have assessed the expression of brown fat markers *Ucp1* and *Elovl3* in subcutaneous adipose tissue from Alk7 knock-out and Ap2CRE conditional mice.

There was no change in subcutaneous fat of conditional mutants and, if anything, a decrease in the knock-outs. Thus, the changes observed in energy balance cannot be due to browning of subcutaneous adipose tissue.

*Chronic administration of the CL compound or cold exposure yields increasing numbers of brite/beige adipocytes in normal mice. The authors should test whether "browning" is enhanced under such protocols*.

We have now done this experiment and report (in panels C and D of new Figure 3—figure supplement 1) that CL injection induced *Ucp1* and *Elovl3* expression to the same extent in subcutaneous adipose tissue of wild type and Alk7 knock-out mice. Thus, the different CL sensitivity of wild type and mutant mice cannot be attributed to differential browning in the mutants.

*3) Metabolic characterization. Previous work by this group demonstrated that Alk7*^*-/-*^
*mice have a metabolic dysfunction but the current manuscript does not include any data testing whether glucose metabolism or insulin sensitivity is perturbed in the new Alk7 mouse models. The authors should perform glucose tolerance tests, insulin tolerance tests, and fasting glucose and insulin measurements*.

We have performed glucose tolerance tests in *Ap2Cre* conditional *Alk7* mutants and controls. The results (shown in panels E, F and G of revised Figure 2) demonstrate comparable fasting glucose levels after chow (zero time point in panel E), and after HFD (zero time point in panel F), comparable insulin fasting levels (zero time point in panel G), and comparable blood glucose and serum insulin levels after glucose injection in control and conditional mutant mice (GTT test).

We have also performed insulin tolerance test in *Ap2Cre* conditional *Alk7* mutants and controls (shown in panel H of revised Figure 2). The result demonstrates normal insulin sensitivity in the mutants. The different insulin baselines in panels G and H are explained by the different fasting conditions used (overnight versus 3h, respectively).